

---

# Holocene sea-level change on the west coast Bohai Bay, China

Fu Wang[1,2], Yongqiang Zong[3], Barbara Mauz[4,5], Jianfen Li[1,2], Jing Fang[6], Lizhu Tian[1,2],

Yongsheng Chen[1,2], Zhiwen Shang[1,2], Xingyu Jiang[1,2], Giorgio Spada[7] and Daniele Melini[8]

[1]Tianjin Center of Geological Survey, China Geological Survey (CGS), Tianjin, China

[2]Key Laboratory of Coast Geo-Environment, China Geological Survey, CGS, Tianjin, China

[3]Department of Earth Sciences, The University of Hong Kong, Hong Kong SAR, China

[4]School of Environmental Sciences, University of Liverpool, Liverpool, UK

[5]Department of Geography and Geology, University of Salzburg, Salzburg, Austria

[6]College of Urban and Environmental Science, Tianjin Normal University, Tianjin, China

[7]Department of Science, University of Urbino, Urbino, Italy

[8]Istituto Nazionale di Geofisica e Vulcanologia, Roma, Italy

*Correspondence to to: Fu Wang (wfu@cgs.cn)*

Abstract. To constrain models on global sea-level change regional proxy data on coastal change are

indispensable. Here, we reconstruct the Holocene sea-level history of the northernmost East China Sea shelf.

This region is of great interest owing to its apparent far-field position during the late Quaternary, its broad shelf

and its enormous sediment load supplied by the Yellow River. This study collected 15 sediment cores from the

coastal plain of west Bohai Bay and extracted 25 sea-level index points through the analyses of sedimentary

facies, foraminiferal assemblages and radiocarbon dating. These proxy data indicate a phase of rapid rise from

c. -17 m to -4 m of mean sea level between c. 10 ka and 6.5 ka. This was followed by a phase of slow rise from

6.5 ka to 2 ka. In contrast to previous studies our data suggest that the sea level remained c. 2.5 - 1 m below

the modern mean sea level during the mid-late Holocene. The difference between proxy data and sea-level

predictions based on three GIA models suggests that the Bohai coastal plain experiences subsidence at a rate

of around 1.25 mm/a since about 7 ka which masks the mid-Holocene highstand recorded elsewhere in the

region. Thus, during the early Holocene rapid rise the sea flooded the coastal plain and the shoreline retreated

landwards at a rate of c. 40 m/a. It stayed at the landward maximum marine limit during the mid Holocene

when the sea-level rise slowed down allowing vertical sedimentary accretion to occur in the landward areas.



During the late Holocene fluvial sediment supply outpaced the sea-level change and the shoreline prograded
seawards at a rate between 20 and 10 m/a.
KEYWORDS: Sea level; Holocene; Glacial Isostatic Adjustment; Ice Equivalent Sea Level; Bohai Bay
**1.    Introduction**
The sea-level rise since the mid-19th century is one of the major challenges to humanity of the 21st
century (IPCC, 2014). The driving mechanisms of this rise are relatively well-known on a global scale, but
on a regional scale the mechanisms are modified by local parameters. One of these parameters is the regional
Holocene sea-level history, which is a background sea-level signal of variable amplitude. In fact, the regional
response to sea-level changes may be very different from the global signal (Nicholls and Cazenave, 2010)
and, understanding regional costal environment is a rising demand of policy makers. Here, we study the
Holocene sea-level history of Bohai Sea, which is the northernmost part of the Yellow Sea. The area is of
special interest because its shoreline is situated on the broad shelf of the East China Sea in the far-field of the
former ice sheets. While the far-field position should allow approximating the ice-equivalent sea level, the
broad shelf is thought to affect the sea-level (Peltier and Dummond, 2002) by up to 10 m (e.g. Milne and
Mitrovica, 2008) in a spatially complex manner. In addition, the exceptionally high supply of fine-grained
fluvial sediment to the bay should have influenced shoreline migration in the past. In order to reliably
constrain the sea level history in such complex settings, high-resolution proxy data are required and
compared with glacio-isostatic adjustment (GIA) model predictions where the difference between model and
proxy datum should allow inferring the non-GIA, hence local, impact on the sea-level history. Our study
builds on earlier work on late Quaternary stratigraphy and coastal evolution (*e.g.* Cang *et al.*, 1979; Geng,
1981; Wang *et al.*, 1981; Wang, 1982; Yang and Chen, 1985; Zhang *et al.*, 1989; Zhao *et al.*, 1978; Fig. 1)
and on published sea-level index points (SLIP) using chenier ridges (Su et al., 2011) and oyster reef (Wang et
al., 2011) as sea-level indicators. Li et al. (2015) draw a sea-level band for the west Bohai Bay based on 136
SLIPs and limiting points (LPs). However, misfits between model outputs and observational data are



apparent (e.g. Wang et al., 2012; Bradley et al., 2016; Li et al., 2015), which are likely caused by the poor
quality of the mid-late Holocene sea-level data and insufficient data for the early-mid Holocene.
**2.      The study area**
The study area lies in a mid-latitude, temperate climate zone (Fig. 1a) on the northwestern coast of the
East China Sea's wide shelf. Geologically, the Bohai Bay is a depression filled by several kilometer-thick
Cenozoic sediment sequences with the top 500 m ascribed to the Quaternary (Wang and Li, 1983). The long-
term tectonic subsidence has been estimated to about 1.3-2.0 mm/a at Tianjin City (Wang et al., 2003). The
Bay is a semi-enclosed marine environment, connected to the Pacific through a gap between the two
peninsulas, Liaodong Peninsulas and Shangdong Peninsulas and the Yellow Sea (Fig. 1b). Our study area is
the central coast of the Bay which lies between two deltaic plains, the Yellow River delta in the south and the
Luan River delta in the north (Fig. 1b). Several small rivers (e.g. Haihe and Duliujianhe, Fig. 1c) cut through
the coastal plain and enter the Bay. the coastal lowland is characterised not only by its low-lying nature,
(less than 10 m above sea level), but also by a series of Chenier ridges situated south of the Haihe River and
buried oyster reefs situated north of the Haihe River (Fig. 1c). Local reference tidal levels such as mean high
waters (MHW) and highest high waters (HHW) are 1.25 m and 2.30 m respectively, based on the four tidal
stations on the west coast of Bohai Bay (Fig. 1c). During the Last Glacial Maximum the shoreline moved to
the shelf break of the Yellow Sea, more than 1000 km to the east and southeast of our study area (e.g. He,
2006). During the Holocene the sea inundated the coastal area with the shoreline moving about 80 km inland
(e.g. Wang et al., 2015). Over 130 SLIPs established for the past 6000 calendar years (e.g. Li et al., 2015)
from the oyster reefs and chenier ridges fall into a band between 2.5 m and -2.5 m elevation.



**3. Methods**
**3.1 Sampling and elevation measurements**
To obtain sedimentary sequences for this study, we consulted previous studies (*e.g.* Cang *et al.*, 1979; Geng,
1981; Wang *et al.*, 1981; Wang, 1982; Yang and Chen, 1985; Zhang *et al.*, 1989; Zhao *et al.*, 1978; Xue et
al., 1993) to learn where in the bay marine deposits are dominant and where the landward limit of the last
marine transgression should occur. We then collected 15 cores along W-E transects from the modern
shoreline to 80 km inland (Fig. 1c), using a rotary drilling corer. Transect A, comprising 6 cores, stretches
from the modern shoreline 80 km inland and crosses the inferred Holocene transgression limit (Xue, 1993).
Transects B, C and D, comprising 9 cores, cross the transgression limit a little further south (Fig. 1c). The
surface elevations of the drilled cores were leveled to the National Yellow Sea 85 datum (or mean sea level,
MSL) using a GPS-RTK system with a precision of 3 cm. The GPS-RTK raw data were corrected and
processed to National Yellow Sea 85 datum system by the CORS system network available from the Hebei
Institute of Surveying and Mapping with National measurement qualification.
**3.2 Sediment and peat analyses**
In the laboratory, the sediment cores were opened, photographed and recorded for sedimentary characteristics
including grain size, color, physical sedimentary structures, and content of organic material. To study the
degree of marine influence in the muddy sediment sequences, sub-samples were collected in 20 cm intervals.
These were analysed with respect to diatoms and foraminifera with a subsequent focus on the foraminifera
due to poor preservation of diatoms. The foraminifera of the >63µm fraction of 20 g dry sample were
counted (e.g., Wang et al., 1985) following studies on modern foraminifera (e.g. Li, 1985; Li et al., 2009).
Sediment description followed Shennan et al. (2015): where in the sediment sequences foraminifera first
appear and/or significantly increase (from zero or less than 10 to more than 50) is noted as transgressive
contact, while the sediment horizon where foraminifera disappear and/or decrease significantly are noted as
regressive contact. These changes are often associated with lithological changes, such as from salt-marsh
peaty sediment to estuarine sandy sediment or tidal muddy sediment across a transgressive contact, or vice



versa. In addition, peat material was analysed in terms of its foraminifera content so that salt-marsh peat can
be differentiated from freshwater peat.
**3.3   Analysis of compaction**
Because the Holocene marine deposits are mainly unconsolidated clayey silt with around 0.74% organic
matter (Wang et al. 2015) post-depositional auto-compaction (Brain et al., 2015) may have led to lowering of
the SLIP. According to Feng et al. (1999), the water content and compaction of marine sediments show
positive correlation with the down-core reduction of water content of the Holocene marine sediment being
about 10%. Based on these observations, we assumed the maximum lowering is about 10% of the total
thickness of the compressible sediment beneath each SLIP. Consequently, the total lowering for an affected
SLIP is 10% of the total thickness of the compressible sequence beneath the dated layer divided by the post-
depositional lapse time proportional to the past 9000 years (e.g. Xiong et al., 2018), i.e. since the marine
transgression in the study area.
**3.4   Radiocarbon analyses**
bulk organic sediment samples and corresponding peat or plant subsamples from salt-marsh peat were
chosen for AMS radiocarbon analysis because these can give more reliable ages for the SLIPs. The resulting
radiocarbon ages were converted to conventional ages after isotopic fractionation were corrected based on
$\delta^{13}C$ results. The conventional radiocarbon ages were calibrated to calendar years using the data set Intcal13
of Calib Rev 7.0.2 for organic samples, peat and plant samples (Reimer, *et al*., 2013). Because Shang et al.
(2018) reported age overestimation of 467 years for the bulk organic fraction of salt-marsh peaty clay
compared to the corresponding peat fraction, all the AMS [14]C ages between 4000 to 9000 BP obtained from
salt-marsh samples were corrected by Y=0.99$X$–466.5 (Y is the corrected age, $X$ is the age obtained from the
organic fractions; Shang et al., 2018) except one <600 years age from borehole Q7 (Table 1).





### 3.5 Sea-level index points (SLIPs)

For determining SLIPs salt-marsh peaty clay layers were used. To convert the dated peat layers into a SLIP,

the modern analogue approach was used by measuring the elevation of the modern open tidal flat (Fig. 2) and

sampling its surface for their foraminiferal content. Following the studies of the modern foraminifera

assemblage (Li, 2009) *Ammonia beccarii* typically occurs in the upper part of an intertidal zone and

*Elphidium simplex* in the lower intertidal zone. The elevation data of the modern analogue samples for which

the foraminifera assemblage confirmed the salt-marsh origin of the peat (i.e. dominance of *Ammonia*

*beccarii*) were then used to infer the indicative meaning of the dated peat layer: the palaeo-mean sea level is

the midpoint between high water of spring tides (HHW:+2.3 m) and mean high waters (MHW:+1.25 m)

which is 1.78 m with ±0.53 m uncertainty (Wang et al., 2002, 2003; Li et al., 2015). For each dated salt-

marsh peat layer the indicative meaning and range, the total amount of possible lowering in elevation due to

sediment compaction and the reconstructed elevation of palaeo-MSL are listed in Table 1.

### 3.6 GIA modelling

The time-evolution of sea level was obtained using the open source program SELEN (Spada and Stocchi,

2007) to solve the "Sea Level Equation" (SLE) in the standard form proposed in the seminal work of Farrell

and Clark (1976). In its most recent development, SELEN (version 4) solves a generalized SLE that accounts

for the horizontal migration of the shoreline in response to sea-level rise, for the transition from grounded to

floating ice and for Earth's rotational feedback on sea level (Spada and Melini, 2019). The programme

combines the two basic elements of GIA modelling (Earth's rheological profile and ice melting history since

the Last Glacial Maximum) assuming a Maxwell viscoelastic incompressible rheology. The GIA models

adopted are ICE-5G(VM2) (Peltier et al., 2004), ICE-6G(VM5a) (Peltier et al., 2012), both available on the

home page of WR Peltier, and the one developed by Kurt Lambeck and colleagues (National Australian

University, denoted as ANU hereafter; Nakada and Lambeck, 1987, Lambeck et al., 2003) provided to us by

A Purcell (pers. com. 2016). Intrinsic uncertainties are estimated from the comparison of GIA predictions

obtained with the models listed above (Melini and Spada, 2019). Table S1 summarises the values used for



each model. The paleo-topography has been solved iteratively, using the present-day global relief given by
model ETOPO1 (Amante and Eakins, 2009). All the fields have been expanded to harmonic degree 512, on
an equal-area icosahedron-based grid (Tegmark, 1996) with a uniform resolution of ~20 km. The rotational
effect on sea-level change has been taken into account by adopting the "revised rotational theory" (Wahr and
Mitrovica, 2011).
**4.    Results**
Lithostratigraphically, the cores show a succession of terrigenous (including fresh-water swamp, river
channel, flood plain), salt-marsh and marine sediments (Table S2) with a clear W-E trend from terrestrial to
marine dominance of deposits (Fig. 3-6). The around 80 km long transect A shows this trend: close to the
modern shoreline pre-Holocene terrigenous sediments are overlain by basal peat including salt-marsh peat or
peaty clay. Further inland these are replaced by fresh-water peat overlain by salt marsh and intertidal
sediments and, above, by terrigenous sediments. The cores DC01, CZ01 and CZ02 are composed of fluvial
sediments only, roughly confirming the Holocene maximum transgression inferred by Xue (1993). Multiple
shifts between salt marsh, marine and fluvial deposits are noticeable in cores QX02, QX03, CZ61 which
originate from the central part of the study area.
Marsh deposits are either a blackish and thin freshwater peat mostly interbedded in yellowish fluvial
sediments or a yellowish-brown salt-marsh peat bearing intertidal foraminifera (Table 1). Their lower
boundaries are usually sharp, and their upper boundaries are mostly diffused or the salt-marsh peat changes
gradually into dark grey intertidal sediments. Salt-marsh peat is intercalated in marine sediment sequences
(i.e. QX01, QX02, CZ61, CZ85, CZ66 and CZ03, Fig. 3-6), particularly at sites that are close to the
Holocene maximum landward limit.
In the core deposits we found *Ammonia beccarii*, *Quinqueloculina akneriana rotunda*, *Protelphidium*
*tuberculatum*. The foraminifera assemblages of the lower part of the intertidal zone and the near-shore
shallow sea area are similar. The abundance is either biased towards *Ammonia beccarii* or it is relatively
small. The latter is most probably due to the area being situated above the MHW and, hence, subject to





evaporation during low tide, with the consequence of a relatively high and highly variable salt content of the
pore water in the intertidal zone. The bias towards salt-tolerant species is confirmed by the modern analogue
samples (Fig. 2, Table 1).
The age of the basal peat ranges between 10047 cal BP and 7829 cal BP (Table 1). The spatial distribution of
the ages confirms the E-W trend of the Holocene transgression where the oldest age is close to the modern
shoreline and the youngest age is close to the maximum transgression limit.
**4.1 Indicative meaning and range**
The data obtained from the modern analogue shows that the tidal flat can be divided into two sub-
environments: intertidal with bioturbation (worm hole developed to tidal surface) and supratidal with salt-
marsh vegetation (Fig. 2). Within the supratidal and salt-marsh zones, the foraminiferal assemblages are
dominated by *Ammonia beccarii* and other intertidal species (Fig. 5) covering an elevational range from
+1.42 m to +2.00 m above msl, including the +1.79m boundary of salt marsh with plants. At sites below
these elevations, i.e. intertidal with bioturbation (Fig. 2), the foraminiferal assemblages are dominated by
*Elphidium simplex*, *Ammonia beccarii and Pseudogyroidina Sinensis*. This foraminiferal zone covers an
elevational range from 1.42 m to the present MSL. Our results from the core sediments show that the
foraminiferal assemblages are mostly dominated by *Ammonia beccarii*.
**4.2 Sea-level Index points**
In total 25 sea-level index points were established from the dated basal salt-marsh peat using the information
obtained from the modern analogue. In Core Q7, at the most seaward location in the study area, the basal
SLIP is dated to ~9700 cal BP (Table 1), marking the onset of marine inundation of the study area. The
overlying marine sequence is capped by a thick layer of shelly gravels at 1.30 m depth and the associated
SLIP is dated to 540 cal BP. This marks the upper end of the marine sequence as foraminifera start to
disappear alongside a change from intertidal to supratidal environmental conditions. The cores ZW15, QX02,
QX03, QX01 show the same sequence as Q7 and provide 6 SLIPs. 19 SLIPs were collected from other cores
(Table 1).

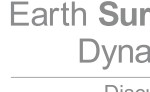

Earth **Surface**
**Dynamics**
Discussions

**4.3 Observed and predicted sea level**
Figure 7a compares observational data and sea-level predictions generated in this study. It shows that none of
GIA models approximates the observations. The difference ranges between around 14 m at 9 ka and 3 m at
2.5 ka. While SLIP data suggest a rising rate of ~0.4 cm/a during the early Holocene, the GIA models
indicate ~0.5 cm/a (ICE-X) and ~0.9 cm/a (ANU). For the mid-late Holocene SLIP data suggest ~0.04 cm/a
rising rate while the GIA models indicate a falling sea level. Predictions obtained from ICE-5G and ICE-6G
are relatively similar but deviate from each other in the timing of the mid-Holocene sea-level highstand. All
three GIA models predict a mid-Holocene sea-level highstand (4.6 m -3.4 m) at 7-6 ka while the SLIP data
remain below modern sea level until 2 ka.
**5. Discussion**
**5.1 Quality of SLIP data**
Owing to elevated salinity of the coastal water samples from both cores and modern tidal flat are
characterised by low microfauna diversity and low number of foraminifera species. This precludes the use of
transfer function statistics and compels analysis based on direct comparison with the modern environment.
We have solved this analytical problem by establishing SLIPs exclusively from basal salt-marsh peats in
transgressive contact and corrected for compaction. With a general uncertainty of around 1.1 m our new
SLIPs are therefore more precise than previously published data (Li et al., 2015) mostly obtained from
chenier ridges, oyster reefs and marine shells. Notwithstanding SLIP improvement in terms of accuracy and
precision, fluctuation of the data exist that can exceed 1 m (e.g. at 3.9 ka and at 5.2 ka, Fig. 7). Although hard
to prove due to lack of data, we believe that these fluctuations are caused by groundwater extraction which
lowers the surface in places.
**5.2 The observed Holocene sea-level rise**
The SLIPs established indicate two phases of sea-level rise during the Holocene. The first phase occurred in
the early Holocene until ~6.5 ka when the sea level rose from -17 m to -4 m. The second phase occurred from



~6.5 ka to 2 ka when the sea level rose from -4 m to -2 m. The oldest Holocene shoreline in Bohai Bay,
situated at -17.2 m and dated to 9700 cal BP, is associated with a transgressive systems tract (Tian et al.,
2017), the water depth of which suggests ~-20 m at 9400 cal BP as the start of the transgression. The
discrepancy is caused by sea ingression occurring ~300 years before Bohai Bay shelf experienced inundation
as indicated by a paleo-river channel deposit (Fig. 2) underlying the Holocene basal peat in core Q7. We take
it therefore for certain that the sea level reached around -15 m at ~9 ka. The final phase from 2 ka to today is
constrained by only one SLLP from core Q7 dated to 540 cal BP at ~0.5 m (Table 1). Lithostratigraphic data
(Shang et al., 2016) suggest that surface of the intertidal sediment body remained very close to zero m from
the landward limit of the marine transgression to about 2 km inland from the present shoreline. Further
inland, in borehole ZW15 the surface elevation of the same intertidal sediment body is ~3.0 m lower than in
core Q7 (Fig. 3, 4) suggesting a rise of sea level in Bohai Bay in the last 1000 years.
**5.3    Observed and predicted Holocene sea level**
We compare our observational data with GIA models employed in this study and with Bradley et al. (2016;
henceforth denoted as BRAD; see also Table S1) who examined several ice-melting scenarios together with a
range of Earth-model parameters, and validated model outputs using published SLIP data from East China
Sea coast including Bohai Bay.
The comparison shows a significant discrepancy for all GIA models (Fig. 7a) including BRAD with
differences ranging between around 14 m in the early-mid Holocene and 3 m in the late Holocene. While the
ICE-X models approximate the observed early Holocene rising rate, the timing of this rise is offset by about
2000 years. In the ANU model the early Holocene sea level rises almost twice as fast as the observed one
with an offset of ~500 years. Thus, the observed sea level rises slower than the modelled sea level. Because
the misfit almost disappears south of Bohai Bay (Fig. S1), the most obvious explanation is subsidence of the
coastal plain. Subsidence is a non-GIA component and should become evident through the residuals (i.e. the
difference between observation and prediction per unit of time; Fig. 7b). Indeed, we identify linearity of
residuals for the period 7-0 ka, suggesting that subsidence dominates the local sea-level signal after the rise


of the eustatic sea level has slowed down. A subsidence rate of 1.25 mm/a is estimated from the residuals,
similar to Wang et al. (2003) who deduced a rate of ~1.5 mm/a from the 400-500 m thick Quaternary
sequence in the bay. It is possible that fluvial sediment supply enhanced the subsidence rate in the Holocene.
The Yellow River's annual discharge into Bohai Bay is estimated to 0.2 Gt until 740AD rising to 1.2 Gt until
around 1800 when widespread farming on the loess plateau started increasing the river's sediment load (Best,
2019). Thus, the sea-level rise in Bohai Bay is in the early Holocene dominated by the global sea-level rise
and associated GIA effects, while in the mid-late Holocene it is dominated by a combination of tectonic
subsidence and fluvial sediment load.
**6.    Conclusions**
Using advanced methods for field survey and identification of sea-level markers, we have established new
precise sea-level index points for the northernmost embayment of the Yellow Sea. Our new data are not only
different to previously published data in that they do not show the expected mid-Holocene sea level
highstand, but also different to global GIA models. We see that as soon as ice melting has ceased, local
processes control shoreline migration and coast evolution. This indicates that more emphasis should be
placed on regional coast and sea-level change modelling under a global sea-level rising future as the local
government need more specific and effective advice to deal with coastal flooding.
**7.    ACKNOWLEDGMENTS**
We thank anonymous reviewers, who gave many constructive suggestions, and this work was supported by
the the China Geological Survey, CGS (DD20189506) and National Natural Science Foundation of China
(Grant no. 41476074, 41806109, 41972196). We also thank Mr. Pei Y. and Shang Z., who contributed to
fieldwork and sample preparation respectively.



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





Earth **Surface** **Dynamics** Discussions — Open Access / EGU


**Table 1. Analytical data used to establish SLIPs.**

| Beta-lab code | Depth (m) | Altitude (m, msl) | Dated material | δ¹³C (‰) | Conventional age (BP) | Calibrated age (BP) (2σ) | Median age (BP) | Indicative meaning and range | Sediment compaction (m)* | Palaeo-mean sea level |
|---|---|---|---|---|---|---|---|---|---|---|
| **Core DC01** | | | | | | | | | | |
| 329636 | 8.40 | -4.66 | Peat | -26.8 | 6950±40 | 7523-7430 | 7487 | Terrestrial peat | | |
| 329637 | 9.27 | -5.53 | Bulk organic | -18.2 | 7410±60 | 8372-8153 | 8248 | Terrestrial peat | | |
| **Core QX01** | | | | | | | | | | |
| 329647 | 5.52 | +0.36 | Bulk organic | -22.5 | 4300±30 | 4892-4829 | 4343** | 1.78±0.53 | 0.29±0.04 | -1.14±0.57 |
| 329644 | 6.35 | -1.19 | Bulk organic | -23.6 | 5010±50 | 5900-5644 | 5226** | 1.78±0.53 | 0.30±0.04 | -2.68±0.57 |
| 329643 | 7.20 | -2.04 | Bulk organic | -25.0 | 5090±30 | 5912-5748 | 5288** | 1.78±0.53 | 0.25±0.03 | -3.58±0.56 |
| 329641 | 8.20 | -3.04 | Peat | -24.6 | 5830±30 | 6732-6554 | 6647 | 1.78±0.53 | 0.24±0.03 | -4.58±0.56 |
| 329642 | 8.70 | -3.54 | Peat | -24.3 | 6030±40 | 6981-6778 | 6875 | 1.78±0.53 | 0.21±0.03 | -5.11±0.56 |
| 329645 | 9.16 | -4.00 | Peat | -27.4 | 6220±40 | 7250-7006 | 7117 | 1.78±0.53 | 0.18±0.02 | -5.60±0.55 |
| 329640 | 11.39 | -6.23 | Peat | -25.3 | 7010±30 | 7935-7786 | 7855 | 1.78±0.53 | 0.01±0.01 | -8.00±0.54 |
| 329646 | 13.05 | -7.89 | Peat | -25.1 | 7200±30 | 8057-7952 | 8002 | Terrestrial peat | | |
| **Core QX03** | | | | | | | | | | |
| 353792 | 2.91 | 1.47 | Peat | -20.6 | 2350±30 | 2461-2326 | 2357 | Terrestrial peat | | |
| 353794 | 4.90 | -0.42 | Peat | -24.0 | 3390±30 | 3699-3569 | 3634 | 1.78±0.53 | 0.16±0.02 | -2.01±0.55 |
| 353796 | 7.39 | -3.01 | Plant material | NA | 5930±30 | 6799-6671 | 6752 | 1.78±0.53 | 0.10±0.02 | -4.68±0.55 |
| 353798 | 8.63 | -4.25 | Plant material | -26.7 | 6410±40 | 7420-7271 | 7350 | 1.78±0.53 | 0.01±0.01 | -6.02±0.54 |
| 353800 | 9.60 | -5.22 | Plant material | -28.2 | 6690±40 | 7622-7478 | 7562 | Terrestrial peat | | |
| 353802 | 12.40 | -8.02 | Plant material | -28.3 | 7280±40 | 8429-8325 | 8397 | Terrestrial peat | | |
| **Core QX02** | | | | | | | | | | |





Earth **Surface**
Dynamics
Discussions

| | | | | | | | | | | |
|---|---|---|---|---|---|---|---|---|---|---|
| 332798 | 3.65 | -0.08 | Bulk organic | -23.6 | 3680±30 | 4091-3913 | 3517** | 1.78±0.53 | 0.30±0.04 | -1.57±0.57 |
| 332792 | 5.68 | -2.11 | Bulk organic | -24.0 | 5450±30 | 6300-6204 | 5718** | 1.78±0.53 | 0.36±0.04 | -3.54±0.57 |
| 333329 | 7.27 | -3.70 | Peat | -26.7 | 6350±30 | 7331-7240 | 7283 | 1.78±0.53 | 0.32±0.04 | -5.16±0.57 |
| 333330 | 8.98 | -5.41 | Peat | -26.3 | 6600±30 | 7522-7434 | 7494 | 1.78±0.53 | 0.19±0.02 | -7.00±0.55 |
| 333331 | 10.97 | -7.40 | Peat | -27.2 | 7020±30 | 7934-7792 | 7867 | Terrestrial peat | | |
| 333333 | 12.42 | -8.85 | Peat | -26.3 | 7140±40 | 8023-7925 | 7966 | Terrestrial peat | | |
| **Core ZW15** | | | | | | | | | | |
| 255821 | 1.6 | 0.03 | Bulk organic | -22.5 | 2930±30 | 3168-2976 | 2584** | 1.78±0.53 | 0.32±0.04 | -1.44±0.57 |
| 356208 | 12.6 | -10.97 | Plant material | -25.0 | 7450±40 | 8358-8186 | 8271 | 1.78±0.53 | 0.00 | -12.75±0.53 |
| 356209 | 13.5 | -11.87 | Plant material | -25.5 | 7640±40 | 8521-8381 | 8430 | Terrestrial peat | | |
| **Core Q7** | | | | | | | | | | |
| 358054 | 1.3 | 2.16 | Bulk organic | -20.4 | 530±30 | 559-510 | 540 | 1.78±0.53 | 0.10±0.02 | +0.49±0.55 |
| 357153 | 17.2 | -13.74 | Plant material | -28.0 | 7990±40 | 9005-8705 | 8868 | 1.78±0.53 | 0.16±0.02 | -15.36±0.55 |
| 357157 | 18.85 | -15.39 | Bulk organic | -24.6 | 9140±40 | 10411-10226 | 9718** | 1.78±0.53 | 0.00 | -17.18±0.53 |
| **Core CZ01** | | | | | | | | | | |
| 395014 | 15.42 | -8.53 | Peat | -27.5 | 8930±40 | 10099-9914 | 10047 | Terrestrial peat | | |
| **Core CZ02** | | | | | | | | | | |
| 395022 | 12.19 | -6.42 | Peat | -23.1 | 7950±30 | 8980-8648 | 8830 | Terrestrial peat | | |
| **Core CZ03** | | | | | | | | | | |
| 395026 | 4.42 | -0.48 | Bulk organic | -24.2 | 2730±30 | 2877-2762 | 2325** | 1.78±0.53 | 0.12±0.02 | -2.15±0.55 |
| 395027 | 6.15 | -2.21 | Peat | -25.1 | 4790±30 | 5593-5470 | 5517 | 1.78±0.53 | 0.19±0.02 | -3.80±0.55 |
| 395028 | 6.54 | -2.57 | Bulk organic | -27.1 | 5830±30 | 6732-6554 | 6114** | 1.78±0.53 | 0.18±0.03 | -4.18±0.56 |
| 395029 | 7.51 | -3.54 | Peat | -26.7 | 6230±30 | 7251-7019 | 7167 | 1.78±0.53 | 0.14±0.02 | -5.19±0.55 |
| 395030 | 9.22 | -5.25 | Peat | -27.3 | 6640±30 | 7576-7468 | 7528 | 1.78±0.53 | 0.01±0.01 | -7.03±0.54 |



| Lab ID | | | Material | δ | Age | Cal range | Median | | Type | | |
|---|---|---|---|---|---|---|---|---|---|---|---|
| 395031 | 9.34 | -5.37 | Peat | -20.0 | 6660±30 | 7583-7483 | 7535 | 1.78±0.53 | | 0.00 | -7.15±0.53 |
| 395032 | 10.23 | -6.26 | Peat | -27.2 | 6900±30 | 7794-7669 | 7726 | Terrestrial peat | | | |
| 395034 | 12.4 | -8.43 | Peat | -27.2 | 7290±30 | 8171-8025 | 8102 | Terrestrial peat | | | |
| **Core CZ87** | | | | | | | | | | | |
| 403413 | 2.66 | 1.8 | Bulk organic | -20.8 | 2420±30 | 2696-2351 | 2446 | Terrestrial peat | | | |
| 403414 | 4.51 | -0.05 | Bulk organic | -23.8 | 3330±30 | 3637-3477 | 3566 | Terrestrial peat | | | |
| 406826 | 5.75 | -1.29 | Bulk organic | -24.1 | 4020±30 | 4536-4420 | 3970** | 1.78±0.53 | | 0.25±0.03 | -2.83±0.56 |
| 403417 | 11.05 | -6.59 | Plant material | -27.9 | 6300±30 | 7275-7165 | 7223 | 1.78±0.53 | | 0.04±0.01 | -8.33±0.54 |
| 403418 | 12.62 | -8.16 | Plant material | -27.6 | 6990±30 | 7876-7736 | 7829 | Terrestrial peat | | | |
| **Core CZ61** | | | | | | | | | | | |
| 407339 | 2.52 | 1.24 | Bulk organic | -20.8 | 2310±30 | 2359-2306 | 2337 | Terrestrial peat | | | |
| 406823 | 4.72 | -0.96 | Plant material | NA | 2780±30 | 2952-2793 | 2877 | 1.78±0.53 | | 0.16±0.02 | -2.58±0.55 |
| 406824 | 6.20 | -2.44 | Bulk organic | -23.9 | 6100±30 | 7029-6884 | 6433** | 1.78±0.53 | | 0.25±0.03 | -3.98±0.56 |
| 403397 | 9.73 | -5.97 | Plant material | -19.6 | 6760±30 | 7664-7577 | 7615 | 1.78±0.53 | | 0.00 | -7.75±0.53 |
| 403398 | 11.04 | -7.37 | Plant material | -27.5 | 7000±30 | 7932-7756 | 7842 | Terrestrial peat | | | |
| 403399 | 12.90 | -9.14 | Plant material | -28.0 | 7160±30 | 8018-7939 | 7980 | Terrestrial peat | | | |
| **Core CZ65** | | | | | | | | | | | |
| 399705 | 4.93 | -1.97 | Bulk organic | -18.5 | 3920±30 | 4428-4280 | 3397 | Terrestrial peat | | | |
| 399708 | 9.58 | -6.62 | Plant material | -27.2 | 7000±30 | 7883-7756 | 7823 | 1.78±0.53 | | 0.01±0.01 | -8.39±0.54 |
| 399710 | 11.50 | -8.54 | Plant material | -27.1 | 7250±30 | 8162-8001 | 8080 | Terrestrial peat | | | |
| **Core CZ80** | | | | | | | | | | | |
| 403401 | 3.73 | 2.69 | Bulk organic | -20.3 | 3170±30 | 3452-3346 | 3400 | Terrestrial peat | | | |
| 403403 | 6.57 | -0.15 | Bulk organic | -22.1 | 5050±30 | 5901-5726 | 5298 | 1.78±0.53 | | 0.20±0.03 | -1.74±0.56 |
| 406825 | 8.75 | -2.33 | Peat | NA | 5840±30 | 6736-6562 | 6660 | 1.78±0.53 | | 0.09±0.01 | -4.02±0.54 |

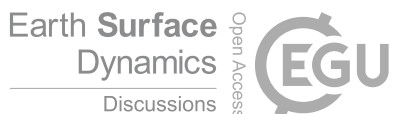

|  |  |  |  |  |  |  |  |  |  |  |
|---|---|---|---|---|---|---|---|---|---|---|
| 403408 | 11.53 | -5.11 | Plant material | -27.5 | 6450±30 | 7428-7313 | 7370 | Terrestrial peat |  |  |
| 403409 | 12.05 | -5.63 | Plant material | -27.9 | 6610±30 | 7565-7440 | 7503 | Terrestrial peat |  |  |
| 403410 | 12.34 | -5.92 | Plant material | -26.4 | 6860±30 | 7759-7618 | 7687 | Terrestrial peat |  |  |
| 403411 | 13.84 | -7.42 | Plant material | -24.6 | 7300±30 | 8175-8029 | 8105 | Terrestrial peat |  |  |
| **Core CZ85** |  |  |  |  |  |  |  |  |  |  |
| 399719 | 3.67 | 0.94 | Bulk organic | -20.5 | 3460±30 | 3671-3641 | 3225** | 1.78±0.53 | 0.17±0.03 | -0.68±0.56 |
| 399720 | 6.77 | -2.16 | Bulk organic | -25.4 | 5830±30 | 6732-6554 | 6114** | 1.78±0.53 | 0.08±0.01 | -3.87±0.54 |
| 399721 | 8.33 | -3.72 | Plant material | -26.4 | 6020±30 | 6947-6785 | 6862 | 1.78±0.53 | 0.01±0.01 | -5.49±0.54 |
| 399722 | 12.70 | -8.09 | Plant material | -28.0 | 7270±30 | 8165-8015 | 8096 | Terrestrial peat |  |  |
| **Core CZ66** |  |  |  |  |  |  |  |  |  |  |
| 399712 | 3.62 | 0.25 | Bulk organic | -23.4 | 3930±30 | 4440-4282 | 3856** | 1.78±0.53 | 0.32±0.04 | -1.22±0.57 |
| 399713 | 5.21 | -1.34 | Bulk organic | -25.1 | 5730±30 | 6632-6445 | 5992** | 1.78±0.53 | 0.39±0.05 | -2.74±0.58 |
| 399714 | 8.14 | -4.27 | Plant material | -27.4 | 6710±30 | 7651-7510 | 7581 | 1.78±0.53 | 0.24±0.03 | -5.81±0.56 |
| 399715 | 10.03 | -6.16 | Plant material | -26.6 | 6790±30 | 7675-7587 | 7635 | 1.78±0.53 | 0.08±0.01 | -7.86±0.54 |
| 399716 | 12.49 | -8.62 | Plant material | -27.1 | 7220±30 | 8156-7965 | 8021 | Terrestrial peat |  |  |
| 399718 | 13.63 | -9.76 | Plant material | -27.6 | 7670±30 | 8523-8406 | 8452 | Terrestrial peat |  |  |

s* Sediment compaction = 10% of compressible thickness divided by lapse time of deposition in the past 9000 years
** corrected for marine influence on salt marsh organic sample fraction ages of peaty clay

Earth **Surface**
Dynamics
Discussions

**Figure captions**

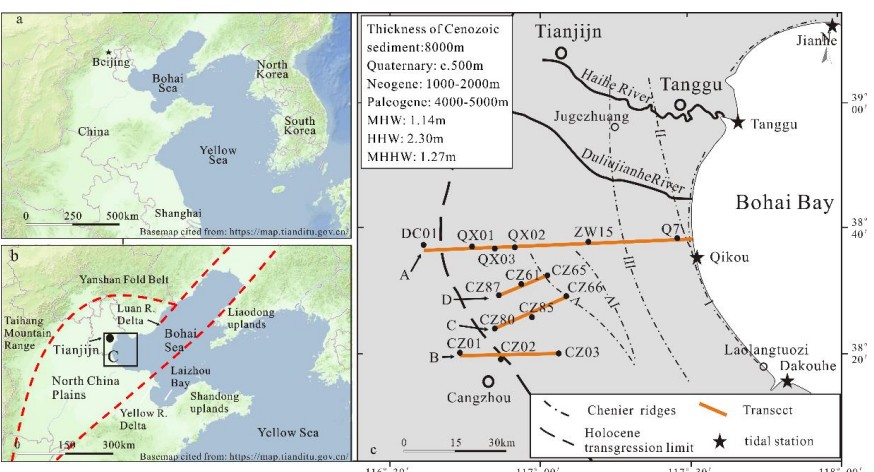


**Figure 1. The study area; (a) location of Bohai Bay and Yellow Sea; (b) location of the study area and major river**
**deltas; red dashed lines indicate the topographic boundaries of coastal lowland, (c) locations of boreholes,**
**transects A, B, C, D, Chenier ridges (Su et al. (2011; Wang et al., 2011) and Holocene transgression limit (Xue,**
**1993). The basemap of Fig.1a and Fig.1b are cited from "map world" (https://www.tianditu.gov.cn/, National**
**Plateform for Common Geispatial Information Services, China)**

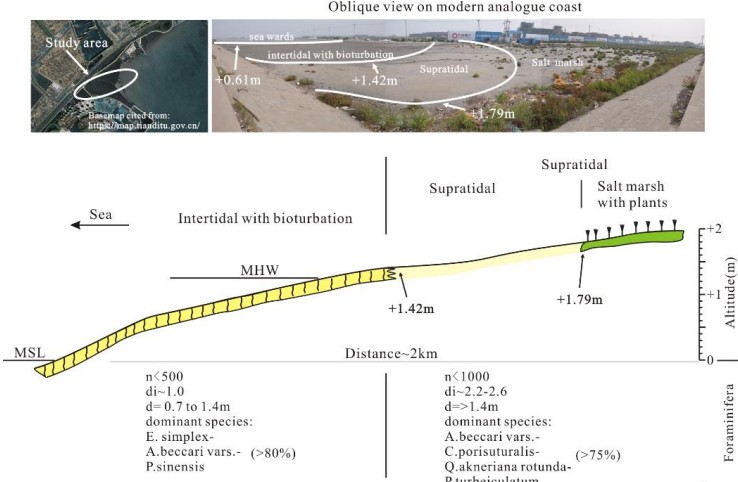


**Figure 2. Schematic cross-section of the modern tidal flat of the study area showing two characteristic**
**foraminiferal zones. The basemap of study area is cited from "map world" (https://www.tianditu.gov.cn/, National**
**Plateform for Common Geispatial Information Services, China)**






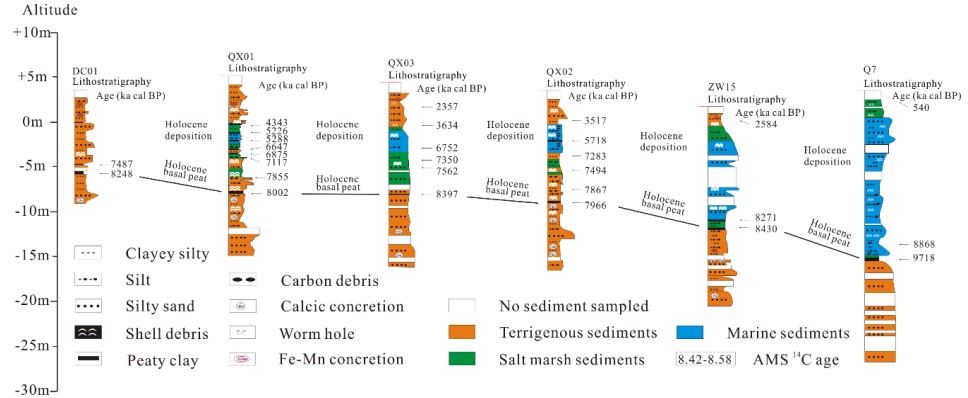


**Figure 3. The lithostratigraphy of transect A, with details of dated sedimentary horizons.**

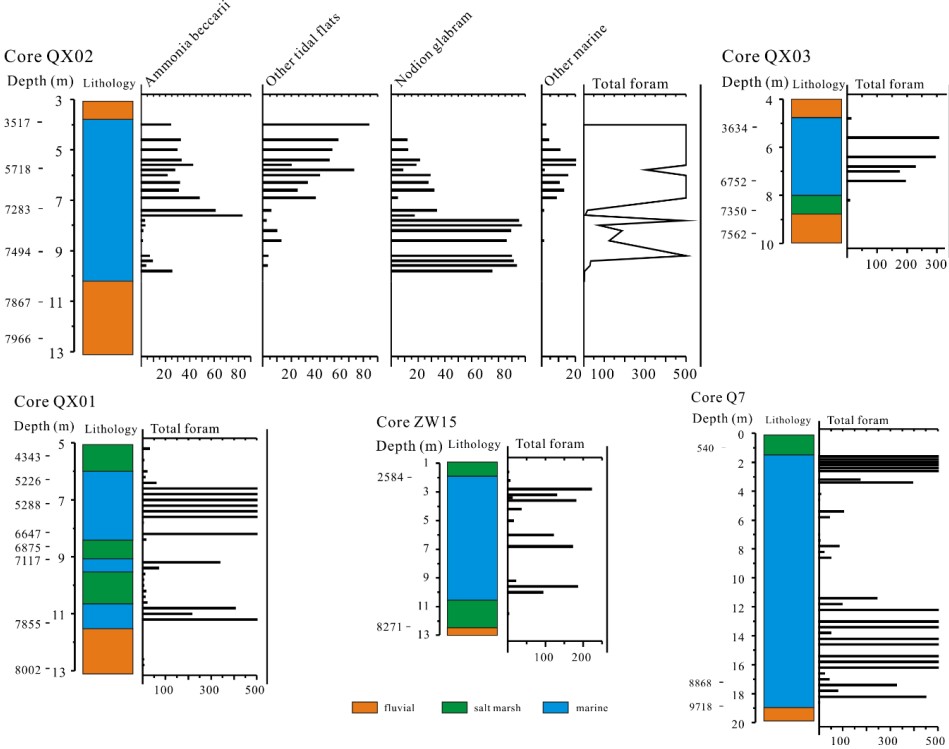


**Figure 4. Foraminiferal counts from five cores of transect A. Counts > 500 are shown as 500.**



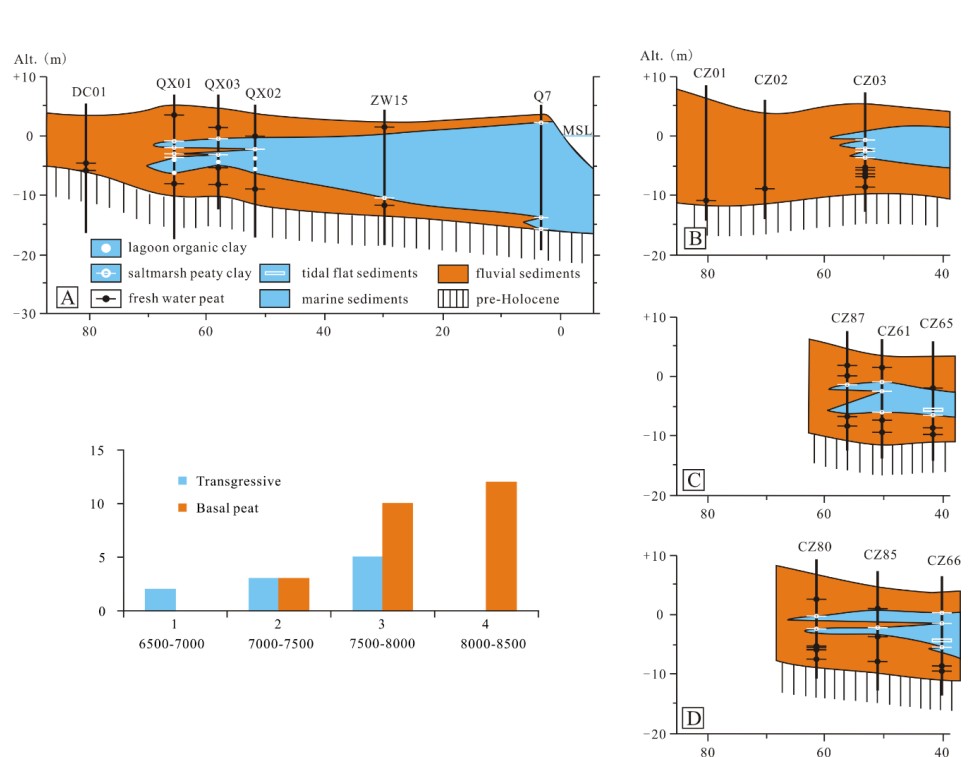


**Figure 5. The lithostratigraphy of transects B, C and D, with details of dated sedimentary horizons.**



Earth **Surface** Dynamics

Discussions

**Core CZ61**

Depth (m)  Lithology  Tidal flats  Lower tidal flats  Marine benthic types  Total foram counts

**Core CZ87**

Depth (m)  Lithology  Total foram

**Core CZ65**

Depth (m)  Lithology  Total foram

**Core CZ03**

Depth (m)  Lithology  Tidal flats  Lower tidal flats  Marine benthic types  Todal foeam counts

**Core CZ80**

Depth (m)  Lithology  Total foram

**Core CZ85**

Depth (m)  Lithology  Total foram

**Core CZ66**

Depth (m)  Lithology  Total foram

fluvial   salt marsh   marine


**Figure 6. Foraminiferal counts from five cores of transects B, C and D. Counts > 500 foraminifera are shown as 500.**

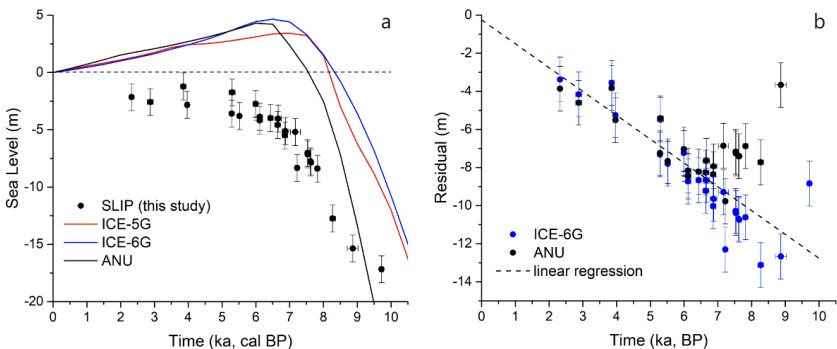

**Figure 7. Observed and predicted sea level in Bohai Bay; (a) SLIPs generated in this study and sea-level predictions. ICE-5G, ICE-6G and ANU are ice models described in section 3.6. For model parameters see Table S1; (b) Sea-level residuals plotted against time. Residuals are the difference between SLIPs and interpolated model data points. Error bars are derived from SLIP uncertainties. The trend line (dashed line) is computed as a least-squares regression on the mean residuals obtained with ANU and ICE-6G. The regression line approximates zero elevation remarkably closely which gives confidence that the calculated 1.25 mm/a for the non-GIA component is correct.**



512 Author contribution

| Author name | Contributions |
| --- | --- |
| Fu Wang | Scientific questions choice, location choice of the boreholes, sampling, measuring, data analyse, results and discussion, entire paper writing. |
| Yongqiang Zong | Revise the part of of the paper and English writing. |
| Barbara Mauz | Revise the part of the paper and English writing. |
| Jianfen Li | Sampling and foraminifera analyse. |
| Jing Fang | Sampling and foraminifera analyse. |
| Lizhu Tian | Sampling and foraminifera analyse. |
| Yongsheng Chen | Sampling and foraminifera analyse. |
| Zhiwen Shang | Sampling and foraminifera analyse. |
| Xingyu Jiang | Sampling and foraminifera analyse. |
| Giorgio Spada | Modelling sea level part "3.6" and "5.3" |
| Daniele Melini | Modelling sea level part "3.6" and "5.3" |

513