# Peer review of "Holocene sea-level change on the central coast of Bohai Bay,"

_Earth Surface Dynamics, 2020_

## Referee Comment (RC1) · Anonymous Referee #1 · 6 May 2020

This manuscript entitled "Holocene sea-level change on the west coast Bohai Bay, China" by Wang et al. reports a number of sea-level index points of the Holocene Bohai Bay area with sound and careful reconstructions. Such a data set is valuable to the scientific community. This manuscript can be accepted after improving the presentation quality. Detailed comments are listed below. In addition to addressing the comments, the authors also need to ask a native English speaker to go through the revised paper to correct and refine written English.

Title: the west coast of Bohai Bay?

Abstract: The shoreline retreat and advance rates are never reported and discussed in the main text. They just pop up here.

[Figure]

L26. The rapid rise of the sea?

Introduction: I suggest authors rewrite this section which is weakly organized. The structure had better come in an order of a brief review of related sea-level studies leading to specific scientific questions targeted by this study and finally the solution of this study.

L33-38 is vague and basically makes no sense.

L41-44. What's the link between these two issues and this study? I did not see they were further discussed. "allow approximating"? It would be clear to say far-field sites usually have a sea-level history similar to IESL. "affect the sea level by up to 10 m" is also an unclear description.

Section 3.4. It is necessary to point out the radiocarbon testing lab to which the samples were sent. The IntCal13 is a dataset independent from Calib program. So, it is not "IntCal13 of Calib". L119, "attained from bulk organic samples"?

L122: Change to "To develop SLIPs, salt-marsh. . ."

L126-128. This sentence is too complicated to follow.

L152-176. Why not set an independent section talking about lithostratigraphy?

L166. Holocene maximum transgression limit?

L169-172 is hard to follow.

Section 4.2 provides limited information of the SLIPs, which need detailed explanation combined with the lithostratigraphy, facies and foram results. More importantly, the reasons why they are chosen and what indicative meaning they are assigned should be given one by one. I suggest reorganizing Section 4. The modern analogue and indicative meaning can be put into new section 4.1. While the lithostratigraphy and SLIPs can be grouped in section 4.2. The comparison of reconstructed and modeled sea-level histories can be moved into the discussion.

L182. There are plus symbols before elevations, so the "above" is redundant.

L221. I have a doubt about making the sample with a date of 9718 cal. a BP in core Q7 as a SLIP. This could be biased by the old carbon effect. In this section, authors need to compare their results with qualified sea-level records from other far-field regions, which were at a much lower height possibly below -25 m in 9700 cal. a BP. Not to mention that the SLIPs from this study have not been calibrated to remove the significant tectonic subsidence.

This conclusion is nearly void.

Figure 1a should be zoomed out to display the entire East China Sea shelf, which also needs a label. Currently, it is similar to figure 1b.

Figure 5. What does the bar chart mean? If the authors want to keep it, it deserves a name like "E". The caption missed the transect A. In addition, please also keep a uniform format for the names of sub-figures. All in upper case or lower case.

Figure 7. Age error bars should be plotted for each SLIP.

---

## Referee Comment (RC2) · Sarah Louise Bradley (Referee) · 11 May 2020

**Review of: Holocene sea-level change on the west coast of Bohai Bay, China.**

The authors have produced 25 new SLIP record from the coast of china which will be a very useful addition to the records from this area. By calculating the residual between the SLIP and predicted relative sea-level from 2 GIA models, they have estimated a subsidence rate of the region and rates of RSL/

I do not have a background in the generation of sea-level data so I will focus my comments related to the section on GIA modelling and general questions.
I have a number of issues with the GIA modelling and the method as I have outlined in the section below which need to be addressed before the paper can be published.

**Missing information:**
I am not sure if the paper is supposed to have SOM material - as it was not uploaded with the main manuscript, but I found a version within the uploaded earlier version.
In this document:
(i) what are the SLIP data on the graph? Is this different from the record in the paper?  It is from a latitude of 33-37N.
(ii) Table: the Bradley model is not shown but is referenced in the paper and is missing.

Given this information, I am reviewing the paper without using this information.

**Abstract:**
There is discussion in the abstract about information that is not provided in the paper:
*Line 26-30*. This reads as a similar to the discussion and results in the Wang, 2015. I would suggest either revising the abstract to be related to what is presented in the paper or adding more work to the paper.
**Introduction:**
I would like the authors to provide an overview on the character of the Holocene sea level across China. As Zong, 2004 (given in the paper) has previously published, there is only a very minor highstand recorded at the other sites across the region. It is therefore important to put your new SLIP record into this context. A SLIP record which does not record a large highstand, as the authors have found is not unusual for the region. The authors do reference some other material, but as I state below it is not in English so unfortunately is not accessible by the wider scientific community to which ESD is aimed.

*Page 3: line 70: 'During the Holocene the sea inundated the coastal area with the shoreline moving 80 km inland' (Wang et al., 2015).*  Can you state how this was determined? From briefly looking at this paper, it describes three of the cores described in this paper. Is this new study a follow on from this?

*Page 3: line 71: Over 130 SLIPs established for the past 6000 calender years (Li et al., 2015)*

Are the authors referring to another study from this region that has already published 130 SLIP? Where are these from? Are they relevant to the paper? If there are 130 extra points from this region - they should be included within this study as this paper is referred to through the study.

**References:**
The authors refer to a number of papers to give support to their statements in the paper (for example, Liu et al., 2015 (as listed above)). Unfortunately as these publications are not in English they are not likely to be accessible to the wider scientific audience. It would be useful if they authors can elaborate at least the method in the papers which the statements they use are based.

**GIA modelling+ Residual: method**
Fig7: List the earth model used in the figure caption.
Page 29: line 507: '*interpolated model data points'.*  Do the authors use just the RSL prediction to take the residual, rather than generating a RSL at the exact location and time of each SLIP?

The authors have looked into the sensitivity to the ice sheet model  in the GIA- but using two different ice sheet reconstruction in the generation of the predictions; ICE6G and ANU, which have very different end to the timing of global melting.
However I am surprising that this makes so little difference to the resultant RSL predictions? can the authors comment on this
(ii) *On Page 6: line 138: 'Intrinsic uncertainties are estimated from GIA predictions with the models listed above'.* This is not the case and the results do not support this statement,
The authors do not consider the uncertainty to earth model. They reference in the paper, that Bradley et al.  2016 examined a range of earth model and concluded a very different range of earth model parameters were required to resolve the large overprediction to the SLIP data. I am not stating that this is the correct model to use.
However, a some of the ~ 5m overprediction to the SLIP data could be due to inaccurate model parameters. This needs to be considered to support the conclusion the authors draw regarding the rate of subsidence and geography evolution.
As the results in the paper are based on determining the residual between the predicted RSL and the SLIP, this needs to be considered.

**Section 4.2:**
In this section the authors have calculated rates: Please state over which time interval this is for? 9- 6ka have a very different rate c.f 0- 2ka. If the authors are calculating the rate from the data, how are they accounting for the errors?

*Section 5.3: Line 233=237. The results for BRAD model are not provided.*
Line 242: 'Because the misfit disappears in South of Bohai Bay'...*the most obvious explanation is the subsidence of the coastal plain'.*
This is not a correct statement. The misfit does not appear.  I do not disagree that some of this misfit is due to subsidence but with the evidence the authors have provided it is not possible to conclude all the misfits are related to this process.

Line 251: *'.... early Holocene dominated by global sea-level rise and associated GIA effects which in the mid-late Holocene it is dominated by combination of tectonic subsidence.* The authors do not provide the information to support this statement. There has been little discussion of the signal over the mid-late Holocene. What do the authors define as mid-late? (4-0ka?)

---

## Editor Comment (EC1) · Tom Coulthard (Editor) · 12 May 2020

Dear Authors, Both reviews for the paper are now in (there are none outstanding) and though the paper discussion is open until the 28th May - comments only (not reviews) can come in during this time. Therefore, to speed up the review/publication process you are welcome to start work on revising the manuscript.

Both reviews are supportive of the paper - with one suggesting more changes and asking for certain points to be clarified/answered. When you submit your revised manuscript - please indicate clearly where changes have been made (track changes) along with a document indicating what these changes are. Many thanks, Tom Coulthard

[Figure]

**ESurfD**

---

## Author Comment (AC1) · 28 May 2020

[Figure]

**Fig. S1. The comparison of SLIPs generated for the coast south of Bohai Bay (latitude 37°N – 33°N; adopted from Lambeck et al., 2014) and GIA models employed in this study.**

**Table S1. Parameters used by GIA models employed in this study (SELEN) compared the model established by Bradley et al. (2016; BRAD).**

| Parameter/Result | BRAD | ANU (SELEN) | ICE 5G (SELEN) | ICE 6G (SELEN) |
|---|---|---|---|---|
| Lithospheric thickness (km) | 96 | 65 | 90 | 90 |
| Upper mantle viscosity (Pa s) | $<1.5 \times 10^{20}$ | $0.5 \times 10^{21}$ | $0.5 \times 10^{21}$ | $0.5 \times 10^{21}$ |
| Lower mantle viscosity (Pa s) | $8 \times 10^{21}$ | $10 \times 10^{21}$ | $2.7 \times 10^{21}$ | $3.2 \times 10^{21}$ |
| Antarctic contribution to ESL (m) and end of melting (until ka) | 28 until 1 | 30 until 1 | 17.5 until 4 | 13.6 until 4 |
| Holocene highstand (m@ka) | $<0.5$ @ 7 | 4.3 @ 6 | 3.4 @ 7 | 4.7 @ 6.5 |

**Table S2. Survey data and lithostratigraphy of cores.**

| Depth (m) | Alt. (m, asl) | Description |
|---|---|---|
| *Core **DC01** (38°40′09″, 116°39′10″, ground altitude: +3.74 m)* | | |
| 1.0–6.50 | *2.74 to -2.76* | Yellowish brown to grey clayey silt with rusting stains |
| 6.5–9.40 | *-2.76 to -5.66* | Yellowish grey silty clay with black peat layers in various depths |
| 9.4–12.6 | *-5.66 to -8.86* | Brown grey clayey silt with calcium nucleus at base (Pre-Holocene) |
| *Core **QX01** (38°38′52″, 116°48′58″, ground altitude: +5.16 m)* | | |
| 1.0–5.00 | *4.16 to 0.16* | Brown to grey clayey silt with thick laminations and rusting stains |
| 5.0–9.10 | *0.16to -3.94* | Brownish grey clayey silt with thin (5 cm thick) layers of charcoal, organic material and shell fragments in various depths |
| 9.1–11.4 | *-3.94 to -6.24* | Brownish grey clayey silt with small amount of charcoal and shell fragments |
| 11.4–13.8 | *-6.24 to -8.64* | Grey to brownish grey clayey silt laminations, with black peat layers and a sharp contact at upper boundary |
| 13.8–19.6 | *-8.64 to -14.4* | Yellowish brown sandy silt (Pre-Holocene) |
| *Core **QX03** (38°38′52″, 116°53′43″, ground altitude: +4.38 m)* | | |

| Depth (m) | Alt. (m, asl) | Description |
|---|---|---|
| 1.2–4.8 | *3.18 to -0.42* | Dark brown clayey silt with small amount of charcoal. Calcium nucleus and shells in 2.9 – 3.1 m depth |
| 4.8–8.9 | *-0.42 to -4.52* | Dark greyish brown clayey silt with laminations and small amount of shell fragments |
| 8.9–13.7 | *-3.52 to -9.32* | Greyish brown to grey clayey silt with a black peat layer in 12.4-12.5 m depth |
| 13.7–16.0 | *-9.32 to -11.6* | Brown clayey silt, with rusting stains and thick laminations (Pre-Holocene) |

*Core QX02 (38°38′24″, 116°57′24″, ground altitude: +3.57 m)*

| Depth (m) | Alt. (m, asl) | Description |
|---|---|---|
| 1.0–3.90 | *2.57 to -0.33* | Yellowish brown clayey silt with small amount of charcoal |
| 3.9–11.3 | *-0.33 to -7.73* | Brownish grey clayey silt, with shell fragments and rusting stains, and several organic-rich layers |
| 11.3–16.6 | *-7.73 to -13.0* | Yellowish brown clayey silt, with charcoal and fine sand at base, and black peat layers |
| 16.6–20.3 | *-13.0 to -16.7* | Yellowish brown clayey silt, with calcium nucleus developed in various sizes (Pre-Holocene) |

*Core ZW15 (38°40′26″, 117°13′20″, ground altitude: +1.63 m)*

| Depth (m) | Alt. (m, asl) | Description |
|---|---|---|
| 0.8–2.2 | *0.83 to -0.57* | Brown clayey silt, with rusting stains, laminations and an increase in organic matter at 1.60 m of depth |
| 2.2–12.6 | *-0.57 to -10.97* | Greyish brown clayey silt, with small amount of marine shells, laminations throughout |
| 12.6–15.2 | *-10.97 to -13.57* | Grey clayey silt with peat layers at various depths |
| 15.2–17.0 | *-13.57 to -15.37* | Dark yellowish brown clayey silt, with rusting stains and calcium nucleus (Pre-Holocene) |

*Core Q7 (38°39′24″, 117°31′27″, ground altitude: +3.46 m)*

| Depth (m) | Alt. (m, asl) | Description |
|---|---|---|
| 0.0–7.0 | *3.46 to -3.54* | Brown silt, with laminations and marine shells |
| 7.0–18.7 | *-3.54 to -15.24* | Dark grey clayey silt, with shell fragments |
| 18.7–18.9 | *-15.24 to -15.44* | Dark brown peaty clay overlying yellowish brown sandy sediment (the latter as Pre-Holocene) |
| 18.9–25.0 | *-15.44 to -21.54* | Yellowish brown silt sand (Pre-Holocene) |

*Core CZ01 (38°22′29″, 116°46′31″, ground altitude: +6.89 m)*

| Depth (m) | Alt. (m, asl) | Description |
|---|---|---|
| 1.0–6.4 | *5.89 to 0.49* | Dark brown clayey silt, with fine laminations, charcoal, Fe/Mn concretion, and freshwater snails |
| 6.4–15.4 | *0.49 to -8.51* | Dark yellowish brown clayey silt, with rusting stains and calcium nucleus. Black peat layers in various depths |
| 15.8–20.0 | *-8.91 to -13.11* | Very dark greyish brown to very dark grey silt (Pre-Holocene) |

*Core CZ02 (38°21′28″, 116°54′50″, ground altitude: +5.77 m)*

| Depth (m) | Alt. (m, asl) | Description |
|---|---|---|
| 1.0–4.4 | *4.77 to 1.37* | Dark yellowish brown clayey silt, with Fe/Mn concretion. |
| 4.4–15.0 | *1.37 to -9.23* | Brown to light greyish brown silt, with laminations, rusting stains and calcium nucleus. Black peat layers at various depths |
| 15.0–20.0 | *-9.23 to -14.23* | Yellowish brown silt, with rusting stains and calcium nucleus (Pre-Holocene) |

*Core CZ03 (38°22′19″, 117°06′29″, ground altitude: +3.94 m)*

| Depth (m) | Alt. (m, asl) | Description |
|---|---|---|
| 1.0–4.4 | *2.94 to -0.46* | Dark yellowish brown clayey silt, with rusting stains and Fe/Mn concretion |
| 4.4–9.3 | *-0.46 to -5.36* | Dark grey brown clayey silt, with laminations and shell fragments. Organic clay and peat in various depths |
| 9.3–15.0 | *-5.36 to -11.06* | Grey silt with charcoal and two black peat layers |
| 15.0–16.0 | *-11.06 to -12.06* | Yellowish brown sandy silt (Pre-Holocene) |

*Core CZ87 (38°31′39″, 116°54′38″, ground altitude: +4.46 m)*

| Depth (m) | Alt. (m, asl) | Description |
|---|---|---|
| 0.0–5.8 | *4.46 to -1.34* | Light grey to brown clayey silt, with laminations, Fe/Mn concretions and rusting stains |

| Depth (m) | Alt. (m, asl) | Description |
|---|---|---|
| 5.8–11.5 | *-1.34 to -7.04* | Yellowish brown clayey silt, with small amount of shell fragments |
| 11.5–16.0 | *-7.04 to -11.54* | Grey clayey silt, with charcoal, laminations and black peats |
| 16.0–20.0 | *-11.54 to -15.54* | Greyish brown silt (Pre-Holocene) |

*Core CZ61 (38°33′29″, 116°58′50″, ground altitude: +3.76 m)*

| Depth (m) | Alt. (m, asl) | Description |
|---|---|---|
| 0.0–4.5 | *3.76 to -0.74* | Yellowish brown clayey silt and silt, with charcoal |
| 4.5–9.7 | *-0.74 to -5.94* | Brown to grey clayey silt, with marine shell fragments. Organic clay at various depths |
| 9.7–14.7 | *-5.94 to -10.94* | Very dark grey clayey silt, with laminations. Peats in various depths |
| 14.7–18.0 | *-10.94 to -14.24* | Yellowish brown clayey silt, with laminations and small amount of calcium nucleus (Pre-Holocene) |

*Core CZ65 (38°34′47″, 117°04′17″, ground altitude: +2.96 m)*

| Depth (m) | Alt. (m, asl) | Description |
|---|---|---|
| 0.0–3.8 | *2.96 to -0.84* | Dark brown clayey silt, with laminations and charcoal |
| 3.8–9.7 | *-0.84 to -6.74* | Grey silt, with rusting stains, laminations and charcoal in upper and lower ends, marine shells in the middle |
| 9.7–13.8 | *-6.74 to -10.84* | Grey clay and silt, with laminations and charcoal and a black peat layer |
| 13.8–16.6 | *-10.84 to -13.64* | Brown to grey brown clayey silt, with Fe/Mn concretion, calcium nucleus and freshwater snails (Pre-Holocene) |

*Core CZ80 (38°26′12″, 116°53′39″, ground altitude: +6.42 m)*

| Depth (m) | Alt. (m, asl) | Description |
|---|---|---|
| 0.0–5.4 | *6.42 to 1.02* | Light yellowish brown to greyish brown clayey silt, with laminations, rusting stains, charcoals and dark grey peats |
| 5. 4–10.0 | *1.02 to -3.58* | Dark grey clayey silt and fine silt, with organic clay in various depths |
| 10.0–14.0 | *-3.58 to -7.58* | Grey fine silt and clayey silt, with rusting stains and charcoal. Black peats in various depths |
| 14.0–17.0 | *-7.58 to -10.58* | Yellowish brown clayey silt, with laminations in upper layer (Pre-Holocene) |

*Core CZ85 (38°28′09″, 117°01′10″, ground altitude: +4.61 m)*

| Depth (m) | Alt. (m, asl) | Description |
|---|---|---|
| 0.5–3.6 | *4.11 to 1.01* | Dark yellowish brown clayey silt, with plant roots at surface, and charcoals, Fe/Mn concretions in the lower part |
| 3.6–8.8 | *1.01 to -4.19* | Brown clayey silt, with rusting stains, charcoals. Organic clay in various depths |
| 8.8–15.8 | *-4.19 to -11.19* | Dark grey silt, with charcoal and black peats |
| 15.8–17.7 | *-11.19 to -13.09* | Light grey clayey silt, with a few calcium nucleus at base (Pre-Holocene) |

*Core CZ66 (38°31′29″, 117°07′59″, ground altitude: +3.87 m)*

| Depth (m) | Alt. (m, asl) | Description |
|---|---|---|
| 1.0–3.6 | *2.87 to 0.27* | Yellowish brown clayey silt, with Fe/Mn concretion in lower part |
| 3.6–6.3 | *0.27 to -2.43* | Yellowish brown clayey silt, with charcoal and organic clay in various depths |
| 6.3–10.8 | *-2.43 to -6.93* | Yellowish brown silt, with rusting stains and small amount of marine shells |
| 10.8–14.0 | *-6.93 to -10.13* | Light yellowish grey to grey silt and clay, with charcoal and black peats |
| 14.0–16.6 | *-10.13 to -12.73* | Greyish brown clayey silt, with Fe/Mn concretion, freshwater snails and shells (Pre-Holocene) |

---

## Author Comment (AC2) · 28 May 2020

Dear editor Tom Coulthard,

We have started to revised the manuscipt following the comments from the refrees and editor, and will be finished in 7 days. We will upload the revised files as soon as possible.

Best regards, Wang Fu
* * *

---

## Author Response (AR1)

We appreciate the comments obtained from Sarah Bradley and an anonymous referee and have revised the manuscript accordingly.

Here is our reply to the referees' comments in purple.

**Anonymous Referee #1**

Title: the west coast of Bohai Bay?

**Answer:** 'west' was replaced by 'central', please see line1.

Abstract: The shoreline retreat and advance rates are never reported and discussed in the main text. They just pop up here.

**Answer:** this part was deleted, the focus is now on the reconstruction of the Holocene sea level rise in Bohai Bay (line35-40).

L26. The rapid rise of the sea?

**Answer:** changed to 'rapid sea-level rise' (line 25)

Introduction: I suggest authors rewrite this section which is weakly organized. The structure had better come in an order of a brief review of related sea-level studies leading to specific scientific questions targeted by this study and finally the solution of this study.

**Answer**: we feel this is what we did: outline the specific interest of the area due to specific questions (broad shelf effect, far-field, high fluvial sediment supply) of sea-level science which are subsequently explained and referenced. These are key topics of sea level science which we address in our study. We agree that a sentence about our solution was missing – now added (lines 62-63), and a brief review of related sea-level studies were added in "Study area", please see line 81-87.

L33-38 is vague and basically makes no sense.

**Answer:** it is part of the motivation of the study. We feel it makes sense and should stay (line45-50).

L41-44. What's the link between these two issues and this study? I did not see they were further discussed.

**Answer:** We agree and have changed the discussion text (lines277-286) to be in line with the introduction (lines 51-59)

"allow approximating"? It would be clear to say far-field sites usually have a sea-level history similar to IESL. "affect the sea level by up to 10 m" is also an unclear description.

**Answer:** text changed (line 53-55)

Section 3.4. It is necessary to point out the radiocarbon testing lab to which the samples were sent. The IntCal13 is a dataset independent from Calib program. So, it is not "IntCal13 of Calib". L119, "attained from bulk organic samples"?

**Answer:** the dating lab has been added (line 126-128) and Intcal13 separated from software (line 131)

L122: Change to "To develop SLIPs, salt-marsh:"

**Answer:** done (line 138)

L126-128. This sentence is too complicated to follow.

**Answer:** agreed and changed accordingly (lines 142-143)

L152-176. Why not set an independent section talking about lithostratigraphy?
**Answer:** Done, it is now summarised in section 4.1 and listed in Table S2

L166. Holocene maximum transgression limit?
**Answer:** yes and changed (line 182).

L169-172 is hard to follow.
**Answer:** We tried hard to comprehend this comment. We may well be blind but we did not find where the text lacks clarity.

Section 4.2 provides limited information of the SLIPs, which need detailed explanation combined with the lithostratigraphy, facies and foram results. More importantly, the reasons why they are chosen and what indicative meaning they are assigned should be given one by one. I suggest reorganizing Section 4. The modern analogue and indicative meaning can be put into new section 4.1. While the lithostratigraphy and SLIPs can be grouped in section 4.2. The comparison of reconstructed and modeled sea-level histories can be moved into the discussion.
**Answer:** Thank you, yes it was necessary to re-organise the Result section, now done in (1) lithostratigraphy and facies, (2) Foraminifera data (3) Modern analogue and indicative meaning and range, (4) SLIPs.

L182. There are plus symbols before elevations, so the "above" is redundant.
**Answer:** done, please see line 209.

L221. I have a doubt about making the sample with a date of 9718 cal. a BP in core Q7 as a SLIP. This could be biased by the old carbon effect.
In this section, authors need to compare their results with qualified sea-level records from other far-field regions, which were at a much lower height possibly below -25 m in 9700 cal. a BP.
**Answer:** Thank you, this is indeed an important point! We have revised the text accordingly (lines 264-273).
Not to mention that the SLIPs from this study have not been calibrated to remove the significant tectonic subsidence.
**Answer:** Yes, because we wanted to quantify the non-GIA component.

This conclusion is nearly void.
**Answer:** we understand that the referee has different expectations. We have now added the important point about the early Holocene shoreline elevation, please see line302-304 – thank you again for pointing this out to us.

Figure 1a should be zoomed out to display the entire East China Sea shelf, which also needs a label. Currently, it is similar to figure 1b.
**Answer:** Fig.1a has been zoomed out follow this comment

Figure 5. What does the bar chart mean? If the authors want to keep it, it deserves a name like "E". The caption missed the transect A. In addition, please also keep a uniform format for the names of sub-figures. All in upper case or lower case.
**Answer:** the Fig. has been reorganized and rewritten, and the bar chart has been deleted.

Figure 7. Age error bars should be plotted for each SLIP.
**Answer:** age error bars were included in the plot - they are small. A note is now added to the caption.

**Sarah Bradley, Referee #2**
Missing information:
I am not sure if the paper is supposed to have SOM material - as it was not uploaded with the main manuscript, but I found a version within the uploaded earlier version.
**Answer:** Our paper is supposed to have supplement material and we apologise for its missing. The SLIP data on the graph are from Lambeck et al. 2014 data compilation. These data indicate a small (~0.5m) highstand at 7-6 ka occurring the south of Bohai Bay. The data do confirm the Bradley et al model prediction (Bradley et al 2016).

In this document:
(i) what are the SLIP data on the graph? Is this different from the record in the paper? It is from a latitude of 33-37N.
**Answer:** yes, the SLIP data in Fig. S3 are from latitude 33-37N.

(ii) Table: the Bradley model is not shown but is referenced in the paper and is missing. Given this information, I am reviewing the paper without using this information.
**Answer:** We are not sure what you mean with "the Bradley model… is missing" – it is correct that we haven't shown the model result because we don't have the data, but we compared our results with those published in Bradley et al 2016 (Fig. 12, BB)

Abstract:
There is discussion in the abstract about information that is not provided in the paper: Line 26-30. This reads as a similar to the discussion and results in the Wang, 2015. I would suggest either revising the abstract to be related to what is presented in the paper or adding more work to the paper.
**Answer:** agreed and abstract text changed (line 35-40).

Introduction:
I would like the authors to provide an overview on the character of the Holocene sea level across China. As Zong, 2004 (given in the paper) has previously published, there is only a very minor highstand recorded at the other sites across the region. It is therefore important to put your new SLIP record into this context. A SLIP record which does not record a large highstand, as the authors have found is not unusual for the region. The authors do reference some other material, but as I state below it is not in English so unfortunatly is not accessible by the wider scientific community to which ESD is aimed.
**Answer:** a paragraph describing the SLIPs is now added in 'Supplement file', and a brief introduction of previous studies were also added in "Introduction"(line 55-58) and "The study area" (lines81-87).

Page 3: line 70: 'During the Holocene the sea inundated the coastal area with the shoreline moving 80 km inland' (Wang et al., 2015). Can you state how this was determined?
**Answer:** It was determined through the identification of marine layers and their respective elevation. We feel it is not necessary at this point to outline the 'how determined'

From briefly looking at this paper, it describes three of the cores described in this paper. Is this new study a follow on from this?
**Answer:** yes.

Page 3: line 71: Over 130 SLIPs established for the past 6000 calender years (Li et al., 2015) Are the authors referring to another study from this region that has already published 130 SLIP? Where are these from? Are they relevant to the paper? If there are 130 extra points from this region - they should be included within this study as this paper is referred to through the study.
**Answer:** we agree that mentioning Li et al without showing the data points creates confusion. We have now added a paragraph in 'The study area' where the SLIPS are described. In addition, the 136 SLIPs are now displayed in Fig. S2 and the SLIP research in the area is outlined in a supplement paragraph which includes Fig. S1.

References:
The authors refer to a number of papers to give support to their statements in the paper (for example, Li et al., 2015 (as listed above)). Unfortunately as these publications are not in English they are not likely to be accessible to the wider scientific audience. It would be useful if they authors can elaborate at least the method in the papers which the statements they use are based.
**Answer:** see above

GIA modelling+ Residual: method
Fig7: List the earth model used in the figure caption.
**Answer:** done, please see the caption of Fig. 7.

Page 29: line 507: 'interpolated model data points'. Do the authors use just the RSL prediction to take the residual, rather than generating a RSL at the exact location and time of each SLIP?
**Answer:** we have generated RSL curves for a single point that is representative of all the locations. Plotting one curve for each location would give a set of indistinguishable curves, due to inherent long-wavelength character of GIA. Regarding time, we have computed RSL on a grid of time points spaced by 500 years and obtained the RSL predictions at the exact time of each SLIP through linear interpolation.

The authors have looked into the sensitivity to the ice sheet model in the GIA- but using two different ice sheet reconstruction in the generation of the predictions; ICE6G and ANU, which have very different end to the timing of global melting. However I am surprising that this makes so little difference to the resultant RSL predictions?
**Answer:** yes indeed, the models have different histories of melting, but they employ also different lower mantle viscosities. The two models (ICE, ANU) have been developed independently and it is good to see that, at least for our study site, they converge towards a common set of RSL predictions.

can the authors comment on this
(ii) On Page 6: line 138: 'Intrinsic uncertainties are estimated from GIA predictions with the models listed above'. This is not the case and the results do not support this statement, The authors do not consider the uncertainty to earth model.

**Answer:** Thank you for pointing this out. There was a misunderstanding between authors about model uncertainties because the difference between independent model results does reflect model uncertainty, although qualitatively only. We have now deleted the sentence (line 159-160).

They reference in the paper, that Bradley et al. 2016 examined a range of earth model and concluded a very different range of earth model parameters were required to resolve the large overprediction to the SLIP data. I am not stating that this is the correct model to use. However, a some of the ~ 5m overprediction to the SLIP data could be due to inaccurate model parameters. This needs to be considered to support the conclusion the authors draw regarding the rate of subsidence and geography evolution. As the results in the paper are based on determining the residual between the predicted RSL and the SLIP, this needs to be considered.

**Answer:** We understand these concerns. The Bradley et al study was not only part of a PhD research with corresponding resources but also based on >300 SLIP data. Our approach was different: scrutenising the SLIP data and employing "standard" GIA models. This, however, does not allow estimating input uncertainties. Carrying out a comprehensive analysis of model uncertainties associated with input model parameters would well be beyond the scope of this work, also in view of the limited dataset.

Section 4.2:
In this section the authors have calculated rates: Please state over which time interval this is for? 9- 6ka have a very different rate c.f 0-2ka. If the authors are calculating the rate from the data, how are they accounting for the errors?

**Answer:** We think that exact quantification of rates would be misleading. We used the ~ symbol for the rates and 'early Holocene, 'mid-late Holocene' as time intervals. This is to describe the differences and offsets relative to each other within the time intervals which are very obvious from the plot in Fig 7a. We think discussing rough estimates of rates is valid here while (very large) error margins would blur the picture.

Section 5.3: Line 233=237. The results for BRAD model are not provided.

**Answer:** correct and agreed that the text is misleading. We have now separated the comparison between Slips and "our" GIA model results on one side and Slips and Brad model results on the other side (Table S1).

Line 242: 'Because the misfit disappears in South of Bohai Bay'...the most obvious explanation is the subsidence of the coastal plain'. This is not a correct statement. The misfit does not appear. I do not disagree that some of this misfit is due to subsidence but with the evidence the authors have provided it is not possible to conclude all the misfits are related to this process.

**Answer:** yes, agreed and text changed (line 282-286)

Line 251: '.... early Holocene dominated by global sea-level rise and associated GIA effects which in the mid-late Holocene it is dominated by combination of tectonic subsidence. The authors do not provide the information to support this statement.

**Answer:** The support is now provided through the comparison between Bohai Bay SLIPs and SLIPs south of the Bay (Fig. S3). We have also explained the geological structure of the area (lines282-286).

There has been little discussion of the signal over the mid-late Holocene. What do the authors define as midlate? (4-0ka?)

**Answer:** Yes, mid-late is around 4-0 ka (Walker et al 2012, JQS 27, 649).

The last sentence in section 5.3 (now lines 295-296) discusses the mid-late Holocene rise in comparison to the one in the early Holocene. This is indeed not much because we do not have high quality data for the last 2000 years.

[revised manuscript text omitted]

---

## Author Response (AR2)

Dear Editor,

Thanks for the kind comments to this revised manuscript, and we have further revised the manuscript based on the comment of Minor Revise. More detail please see below:

Comment---Minor Revise both reviewers identified requires further attention - the introduction. Reviewer 1 suggests re-writing the introduction - and the track changes for section indicate there are a few but not substantial changes here. Reviewer 2 also asks for a greater overview in the introduction - of which changes have been made in section 2 (study area not introduction) and in the supplemental material.

Therefore - before publication - I request that you revise the introduction - possibly combine it with the study area section (section 2) and provide a clearer context for the work. For the non specialist in the area (and ESurfs readership may be interested but not know all the details of the area and the issues) at present it is not completely clear what the issue you need to investigate is. So please - in the introduction - make sure you cover firstly what the research issue to be addressed is - what the knowledge gap lies. How previous studies have adressed this - and left the issue to be adressed - and finally how this paper aims to address those issues.

**Answer:** We have revise the Introduction, combined the Introduction with section 2 of The study area following the comment, and further revise were carried out, in order to let the Introduction can cover the what the research issue to be addressed is - what the knowledge gap lies. How previous studies have addressed this - and left the issue to be addressed - and finally how this paper aims to address those issues. Please see Introduction section.

Moreover, following the comments, the co-author's affiliations were checked and revised.

[revised manuscript text omitted]